# Larvicidal properties of terpenoid-based nanoemulsions against the dengue vector *Aedes aegypti* L. and their potential toxicity against non-target organism

Jonatas Lobato Duarte[1], Stéphane Duchon[2], Leonardo Delello Di Filippo[1], Marlus Chorilli[1], Vincent Corbel[2,3]*

1 Department of Drugs and Medicines, School of Pharmaceutical Sciences, São Paulo State University (UNESP), Araraquara, São Paulo, Brazil, 2 Institut de Recherche Pour le Développement (IRD), MIVEGEC, CNRS, IRD, Univ. Montpellier, Montpellier, France, 3 Laboratório de Fisiologia e Controle de Artrópodes Vetores (Laficave), Fundação Oswaldo Cruz (FIOCRUZ), Instituto Oswaldo Cruz (IOC), Rio de Janeiro–RJ, Brazil

* vincent.corbel@ird.fr

**Data Availability Statement:** All relevant data are within the paper.

## Abstract

The development of insecticide resistance in mosquitoes of public health importance has encouraged extensive research into innovative vector control methods. Terpenes are the largest among Plants Secondary Metabolites and have been increasingly studied for their potential as insecticidal control agents. Although promising, terpenes are insoluble in water, and they show low residual life which limits their application for vector control. In this study, we developed and evaluated the performances of terpenoid-based nanoemulsions (TNEs) containing myrcene and p-cymene against the dengue vector *Aedes aegypti* and investigated their potential toxicity against non-target organisms. Our results showed that myrcene and p-cymene showed moderate larvicidal activity against mosquito larvae compared to temephos an organophosphate widely used for mosquito control. However, we showed similar efficacy of TNEs against both susceptible and highly insecticide-resistant mosquitoes from French Guyana, hence suggesting an absence of cross-resistance with conventional insecticides. We also showed that TNEs remained effective for up to 45 days in laboratory conditions. The exposure of zebrafish to TNEs triggered behavioral changes in the fish at high doses but they did not alter the normal functioning of zebrafish organs, suggesting a good tolerability of non-target organisms to these molecules. Overall, this study provides new insights into the insecticidal properties and toxicity of terpenes and terpenoid-based formulations and confirms that TNE may offer interesting prospects for mosquito control as part of integrated vector management.

## Introduction

Over the past decade, *Aedes Borne Diseases* (ABDs) such as dengue, zika, chikungunya, and yellow fever have become a global concern due to their emergence or re-emergence [1]. These

**Funding:** This project has received funding from Fundação de Amparo a pesquisa do estado de São Paulo-FAPESP under the grants n°2021/11487-4 and 2019/25125-7. This project also receives a co-funding from the European Union HORIZON EUROPE Marie Sklodowska-Curie - HORIZON-MSCA-2021-SE-01 (INOVEC project), under the grant n°101086257. Views and opinions expressed are however those of the author(s) only and do not necessarily reflect those of the European Union or the European Research Executive Agency (REA). Neither the European Union nor the REA can be held responsible for them. The funders had no role in study design, data collection and analysis, decision to publish, or preparation of the manuscript.

**Competing interests:** The authors have declared that no competing interests exist.

diseases pose a significant threat to over half of the world's population, resulting in thousands of deaths annually, mainly in the Americas, South-East Asia, and Western Pacific regions [2]. Beyond their direct impact on human health and well-being, ABDs may cause tremendous economic impact [3, 4], recently estimated to be US$ 150 billion over the period 1970–2017 [5].

*Aedes aegypti* L. is the primary dengue vector, with a widespread distribution worldwide, a high vectorial capacity, due to its close association with humans in urban settings [6]. *Ae. aegypti* distribution has increased recently due to global changes [7] and it may be more prevalent in temperate regions such as Europe due to the presence of suitable conditions for establishment [8]. The strategies for effectively controlling *Ae. aegypti* essentially rely on environmental management, community participation, and on the use of larviciding and spatial spraying (in case of outbreaks) [9].

As far as chemical control is concerned, organic substances, such as pyrethroids, organophosphates, and to a lesser extent insect growth regulators are mainly used. However, the extensive and repeated use of those chemicals has caused the emergence and spread of insecticide resistant mosquitoes worldwide [10]. For example, recent studies have shown an increasing resistance of *Ae. aegypti* to pyrethroids [11–14] and organophosphates [15–17] due to the presence of target site modifications (eg kdr mutations for pyrethroids) and increased detoxification through the overexpression of P450, GST and esterase genes [18–20]. The occurrence of "super" insecticide-resistant *Ae. aegypti* mosquitoes have even been reported in Southeast Asia due to the presence of 3 different *Kdr* mutations in the same individuals [21]. This rapid and global increase in Aedes resistance makes it more difficult to control field populations with current public health pesticides and may potentially worsen the impact of ABDs worldwide. Moreover, chemicals can be toxic for non-target organisms and the environment [22, 23], and as such, they are facing increasing regulatory constraints and citizen aversions.

Therefore, there's an urgent need for more ecological, durable, and efficient approaches to control ABDs [24]. Natural products show interesting prospects in vector control because they are degraded more easily and are less harmful to the environment than conventional chemicals, and they may contribute to insecticide resistance management in mosquitoes [25]. Among natural products, essential oils (EO) have been extensively studied because they contain suitable active ingredients for the development of plant-based insecticides [26–29]. The monoterpenes represent about 90% of the EO composition and are the main ones responsible for larvicidal and repellent activities [30]. The *P-cymene* and myrcene are hydrocarbon monoterpenes that already showed high potential as larvicidal agents against *Ae. aegypti* [25, 31–34]. However, due to the lipophilic characteristic of the terpenes and their high volatility, they may offer a limited efficacy against mosquitoes in a natural environment.

Consequently, nanoemulsions represent a promising tool for the encapsulation, protection, and improved solubility of lipophilic bioactive components such as terpenes [35, 36]. Nanoemulsions consist of two immiscible liquids and one or more stabilizers, usually surfactant agents, that may ensure better stability and efficacy of these molecules in water environment. Another useful property of nanoemulsions is the controlled release of the molecules; The process of drug release from nanoemulsion entails the movement of the drug from the oil phase to the surfactant layer and subsequently into the aqueous phase. As the solubilized drug diffuses out of the oil, it encounters the surrounding water, leading to nanoprecipitation. This phenomenon significantly increases the drug's surface area [37].

This study aimed to evaluate the effectiveness against insecticide-susceptible and highly resistant *Ae. aegypti* mosquitoes of terpene-based nanoemulsions (TNEs) containing *p*-cymene and myrcene. Biological assays were performed with successive concentrations of p-cymene and myrcene formulations, both alone and in combination, to determine concentration-

**Table 1. Composition of the monoterpenes nanoemulsions.**

|  | Monoterpene (%) | Tween 20 (%) | Span 80 (%) | Water |
|---|---|---|---|---|
| **P-cymene-NE** | 5 | 4.7 | 0.3 | 90 |
| **Myrcene-NE** | 5 | 4.3 | 0.7 | 90 |

NE: Nanoemulsion

response plots and estimate adequate metrics. The study also investigated any cross-resistance mechanism with conventional insecticides that may affect the effectiveness of TNEs. Finally, we assessed the acute toxicity of these nanoemulsions against non-target organisms, specifically Zebrafish.

## Material and methods

### Preparation of formulations

The TENs were obtained by a low-energy method as described in DUARTE *et al*, 2024 [38]. Briefly, an oil phase composed of the terpene (p-cymene or myrcene) (5% w/w) was mixed with the surfactants (Span® 80/Tween® 20) (5% w/w) (Table 1) under a magnetic stirrer and after homogenization, the aqueous phase (90% w/w) was added dropwise. The resulting nanoemulsions had a droplet size of approximately 120 nm and a uniform distribution of particle size. They remained stable for up to 90 days, indicating good colloidal stability.

### Biological material

Two different strains of *Ae. aegypti* were used in the study. The susceptible reference laboratory strain (Bora) originating from French Polynesia has no detectable resistance mechanism [39]. The strain is maintained at IRD for more than 20 years. The Guyana strain has been collected in French Guyana (Ile Royale) and shows high levels of resistance to pyrethroids [40] through the presence of V1016I and F1534C mutations at high frequencies and higher expression of multiple CYP450 genes [18]. The strain is also resistant to organophosphate through higher expression of Esterase genes [40].

### Larvicidal activity and cross-resistance assessment

Larval bioassays were conducted according to the standardized protocol established by the World Health Organization [41]. The protocol involved exposing III-IV instar larvae to different concentrations of terpenes and terpene-based formulations and recording their mortality after 24 hours. The experiments were conducted under controlled conditions, where larvae of *Ae. aegypti* (Bora strain and Guyana strains) were kept at 25 ± 2°C, with a relative humidity of 75 ± 5%, and a light/dark cycle of 12 hours during the test. The experiments were carried out in triplicate, with four replicates of 25 larvae (n = 100) per concentration, as well as a control. Temephos, an organophosphate insecticide commonly used for larval control, was used as the reference product.

Concentration-response curves were determined against both susceptible and pyrethroid resistance colonies of *Ae. aegypti* (Bora and Guyana) in order to assess potential cross-resistance. The resistance ratios ($RR_{50}/RR_{95}$) were determined by comparing the $LC_{50/95}$ obtained on the resistant strain and the susceptible ones. According to WHO $RR_{50} < 5$ indicates a susceptible population; $RR_{50}$ from 5–10 indicates moderate resistance and $RR_{50} > 10$ indicates high resistance [42] In our case, evidence for cross-resistance was demonstrated if resistance ratios ($RR_{50/95}$) exhibited confidence limits excluding the value of 1.

## Interactions between monoterpenes

The existence of synergism between p-cymene and myrcene or Cym-NE and Myr-NE was also investigated by using the combination index (CI) described by Chou and Talalay (1984) [43], using the CalcuSyn software (Chou and Hayball 1996). This isobologram-based method is particularly well adapted to analyse multiple drug effects [44]. The combination index gives a quantitative measure of the interactions (synergism, antagonism, and summation) occurring between two insecticides. A CI = 1, <1, and >1 indicates an additive effect, a synergistic effect, and an antagonistic effect, respectively [45].

## Residual activity of terpene-based formulations

For the residual activity, a stock solution of the TENs was prepared in the concentration of 40 mg/L and stored in a climatic chamber with controlled conditions (temperature: 27˚C and Relative humidity: 80% and a light/dark cycle of 12 hours). Larval bioassays were carried out as previously described using a single dose of 40mg/L at different intervals of time (1, 7 15, 30, and 45 days) and mortality was recorded at 24H post-exposure.

## Toxicity in a non-target organism

**Animals.**   The animals used were adult zebrafish (*Danio rerio*) of the wild AB type, supplied by the company Pisciculture Power Fish, located in Itaguaí-RJ, Brazil. They were maintained in the Zebrafish Platform of the Drug Research Laboratory of the Federal University of Amapá, (UNIFAP- Brazil), with an adaptation period (40 days), a circadian rhythm of 12 hours, (light period from 7:00 a.m. to 7:00 p.m.), with controlled temperature ($23 \pm 2$˚C), and receiving commercial fish food (Alcon Colors, Santa Catarina, Brazil) twice a day [46]. The animal health and behaviour were monitored during all the analysis period (24 hours) and all researchers involved in the experiments have training in animal care. The experiments were authorized by the Ethics Committee on Animal Experimentation of the Federal University of Amapá (Brazil), registration number 006/2021.

**Acute oral toxicity study.**   The acute toxicity evaluation of Cym-NE and Myr-NE in adult zebrafish was determined using the Limit Test as recommended by the Organization for Economic Cooperation and Development (OECD) 425 [47], with adaptations. The animals were separated into treatment groups (5 animals/group), kept fasting for 24 hours, weighed, and treated orally according to the methods described by [48]. Cym-NE was administered at doses of 2000 and 1750 mg/kg and Myr-NE at a dose of 2000 mg/kg, orally by gavage method, with the aid of a volumetric pipette (HTL Lab Solutions Co.), the volume was calculated according to the animal's weight [49].

**Behavioural analysis and mortality.**   After the gavage procedure, the animals were observed for behavioral changes. The observed behavioral changes were classified as Stage I: increase in swimming activity; Spasms; tremors in the tail; Stage II: circular swimming; loss of posture; Stage III: clonus; loss of motility; animal still at the bottom of the aquarium and death [46]. In the absence of a response to mechanical stimulation and the absence of movement of the operculum, the animal was considered dead [46, 49]. At the end of the observations (24 hours) the animals underwent euthanasia in anesthetic cooling, according to the American Guidelines of the Veterinary Medical Association for Animal Euthanasia [50].

**Histopathological analysis.**   For histopathological analysis of the intestine, liver, and kidneys, the animals were fixed in Bouin solution for 24 hours and decalcified in EDTA solution (Tetraacetic Ethylenediamine Acid, Sigma Co., São Paulo, Brazil) for 48 hours. The samples were dehydrated in a series of alcohols (70, 80, 90, and 100%), diaphanized in xylol, and included in paraffin. The samples were sectioned in 5 μm using a microtome (Brand Rotary

Microtome Cut 6062, Slee Medical, Germany), and histopathological analysis was performed after the tissue sections were cordoned with hematoxylin and eosin, as described by Souza et al., (2016) [46]. The images were analyzed under Olympus BX41-Micronal Microscope and photographed with MDCE-5C USB 2.0 (digital) camera.

**Assessment of histopathological changes.** The Index of Histopathological Changes (IHC) was calculated according to the levels of tissue alterations observed in the gills, liver, and kidneys. The changes were classified as levels I, II, and III, and the IHC value indicates whether the organ is healthy (0 to 10), with changes from mild to moderate (11 to 20), with moderate to severe changes (21 and 50) or containing irreversible changes ($>$ 100) [51, 52]. Thus, the indexes were calculated according to the following equation:

$$I = \frac{\sum_{i-1}^{na} ai + 10\sum_{i-1}^{nb} bi + 10^2\sum_{i-1}^{nc} ci}{N}$$

Where: **a**: first stage changes; **b**: second stage changes; **c**: third stage changes; **na**: number of changes considered as the first stage; **nb**: number of changes considered as the second stage; **nc**: number of changes considered as the third stage; **N**: number of fishes analysed per treatment.

## Statistical analysis

For larval bioassays, $LC_{50}$ and $LC_{90}$ values with their 95% confidence intervals were calculated from a log dose–probit mortality regression line using the SPSS software (IBM, USA). The results of acute oral toxicity obtained were expressed as the mean ± standard deviation (SD) of each experimental group. The ANOVA test was applied, followed by the Tukey test. The significance level considered was 5% ($p < 0.05$). The software used was the Prisma® Graph Pad (version 5.03).

## Results

### Larvicidal activity of terpenes and cross-resistance studies

Bioassay data for the susceptible mosquito strain are shown in Table 2 and Fig 1. Temephos (reference product) showed high insecticidal activity against susceptible mosquito larvae, with a $LC_{50}$ and $LC_{99}$ of 0.001 and 0.00177 mg/L, respectively.

On the other hand, myrcene and p-cymene, showed moderate larvicidal activity against the susceptible Bora strain with $LC_{50}$ values of 12.8 and 13.3 mg/L, respectively, and $LC_{99}$ values of 29.5 and 47.9 mg/L, respectively.

**Table 2. Larvicidal activity of monoterpenes, free and in nanoemulsions, against *Ae. aegypti* L. (Bora strain) after 24 hours of exposure using the WHO larval bioassay.**

| | n | $X^2$ | df | *p value* | $LC_{50}$ (mg.L$^{-1}$) (95%CI) | $LC_{99}$ (mg.L$^{-1}$) (95%CI) |
|---|---|---|---|---|---|---|
| Temephos | 3362 | 6.7 | 5 | 0.244 | 0.001 (0.00102–0.00108) | 0.00177 (0.00168–0.00188) |
| Myrcene | 3351 | 20 | 4 | <0.001 | 12.8 (8.8–15.3) | 29.5 (25.0–42.1) |
| Myr-NE | 3352 | 44.7 | 5 | <0.001 | 13.3 (7.1–17.8) | 47.9 (36.9–79.9) |
| P-cymene | 3366 | 188.3 | 5 | 0.003 | 16.0 (14.8–17.0) | 29.4 (26.5–34.3) |
| Cym-NE | 3356 | 9.00 | 4 | 0.061 | 14.0 (13.4–14.7) | 24.7 (22.6–27.8) |

n represents the total number of larvae used. The assays were performed in triplicate. $X^2$ -Chi-square value; df–degree of freedom, $LC_{50-99}$ –Lethal concentration (50 and 99%), 95% CI- 95% Confidence interval. Myr-NE Myrcene Nanoemulsion; Cym-NE; Cymene Nanoemulsion

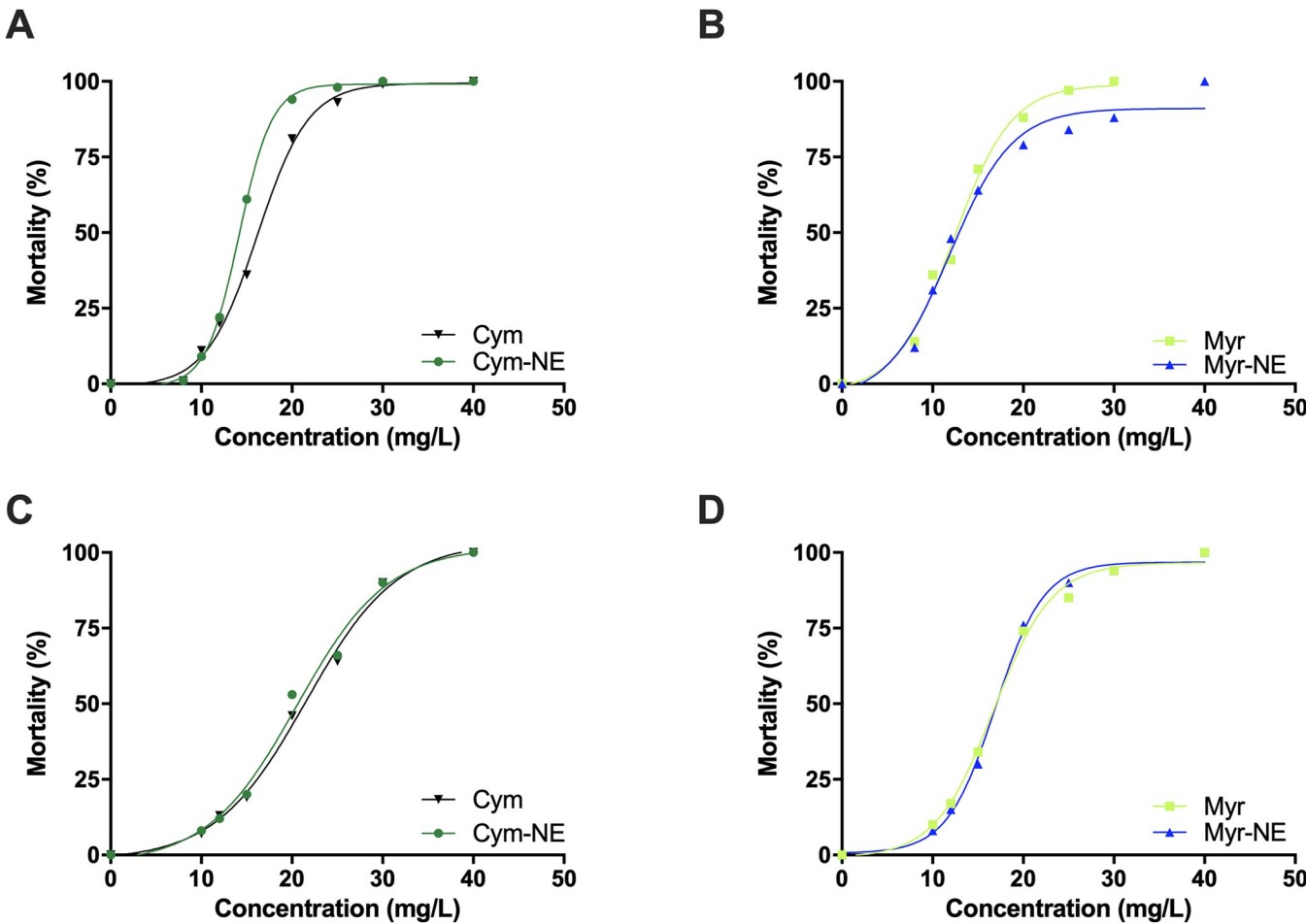

**Fig 1.** Concentration-response plots of terpenes against A*e. Aegypti* larvae: A–Bora strain with Cym and Cym-NE; B–Bora strain with Myr and Myr-NE; C–Guyana strain with Cym and Cym-Ne and D–Guyana strain with Myr and My-NE.

The nanoemulsification process did not significantly affect the overall efficacy of the monoterpenes as $LC_{50}$ and $LC_{99}$ values of Myr-NE and Cym-NE did not differ than the one's of the free monoterpenes (Table 2).

Bioassays data for the resistant mosquito strain is shown in Table 3 and Fig 1. Our finding showed that the Guyana strain was highly resistant to temephos with $RR_{50}$ and $RR_{99}$ of 124

**Table 3. Larvicidal activity of monoterpenes, free and in nanoemulsions against insecticide-resistant *Ae. aegypti* (Guyana strain) after 24 hours of exposure, using the WHO larval bioassay.**

|          | n    | $X^2$ | df | *p value* | $LC_{50}$ (mg.L$^{-1}$) (95% CI) | $LC_{99}$ (mg.L$^{-1}$) (95% CI) | $RR_{50}$ (95% CI) | $RR_{95}$ (95% CI) |
|----------|------|-------|----|-----------|----------------------------------|----------------------------------|--------------------|--------------------|
| Temephos | 2250 | 30.8  | 5  | <0.001    | 0.150 (0.05–0.19)                | 0.450 (0.336–1.743)              | 124.44 (103.40–149.77) | 253.03 (160.90–397.93) |
| Myrcene  | 2000 | 7.8   | 4  | 0.099     | 17.3 (15.0–19.0)                 | 37.8 (33.4–46.2)                 | 1.36 (1.22–1.53)   | 1.30 (1.035–1.64)  |
| Myr-NE   | 2000 | 12.2  | 4  | 0.016     | 17.3 (14.9–19.0)                 | 34.6 (30.3–43.8)                 | 1.25 (1.07–1.47)   | 0.83 (0.59–1.15)   |
| P-cymene | 2000 | 14.5  | 4  | 0.006     | 21.8 (18.8–23.9)                 | 41.8 (35.7–58.9)                 | 1.24 (1.07–1.54)   | 1.55 (1.05–2.29)   |
| Cym-NE   | 2000 | 14.9  | 4  | 0.005     | 21.0 (18.0–23.0)                 | 42.1 (35.8–58.3)                 | 1.39 (1.20–1.60)   | 1.84 (1.34–2.52)   |

n represents the number of larvae used. The assays were performed in triplicate. $X^2$ -Chi-square value; df–degree of freedom, $LC_{50-99}$ –Lethal concentration (50 and 99%), $RR_{50-95}$ –Resistant ration (50 and 95%). Myr-NE Myrcen Nanoemulsion; Cym-NE; Cymene Nanoemulsion

and 253, respectively. Our findings showed no cross-resistance of the Guyana strain to monoterpenes, with $RR_{50}$ and $RR_{95}$ ranging from 0.83 (95%CI 0.59–1.15) to 1.84 (1.34–2.52).Once again, no significant difference in efficacy was observed between free monoterpenes and nanoemulsions against the insecticide resistant *Ae. Aegypti* colony, suggesting that the nanoemulsification process does not impair the biological activity of the p-cymene and myrcene.

### Interactions between monoterpenes

Here, a 1:1 ratio (myrcene:p-cymene) was adopted for the combination considering that both terpenes had similar $LC_{50}$ against both susceptible and resistant mosquitoes. Concentration-response plots of terpenes, alone and in a mixture, are shown in Fig 2.

Our findings showed that the combination of terpenes did not kill a higher proportion of mosquitoes than the sum of the 2 terpenes taken separately. Combination indexes (CI) ranged from 1 to 1.5 hence indicating additive and slight antagonism at high concentrations, respectively according to the classification of CHOU & TALALAY, [43] (Fig 2C and 2D). This indicates that the mortality of the combination at high concentrations is lower than the expected additive effect of the single insecticides.

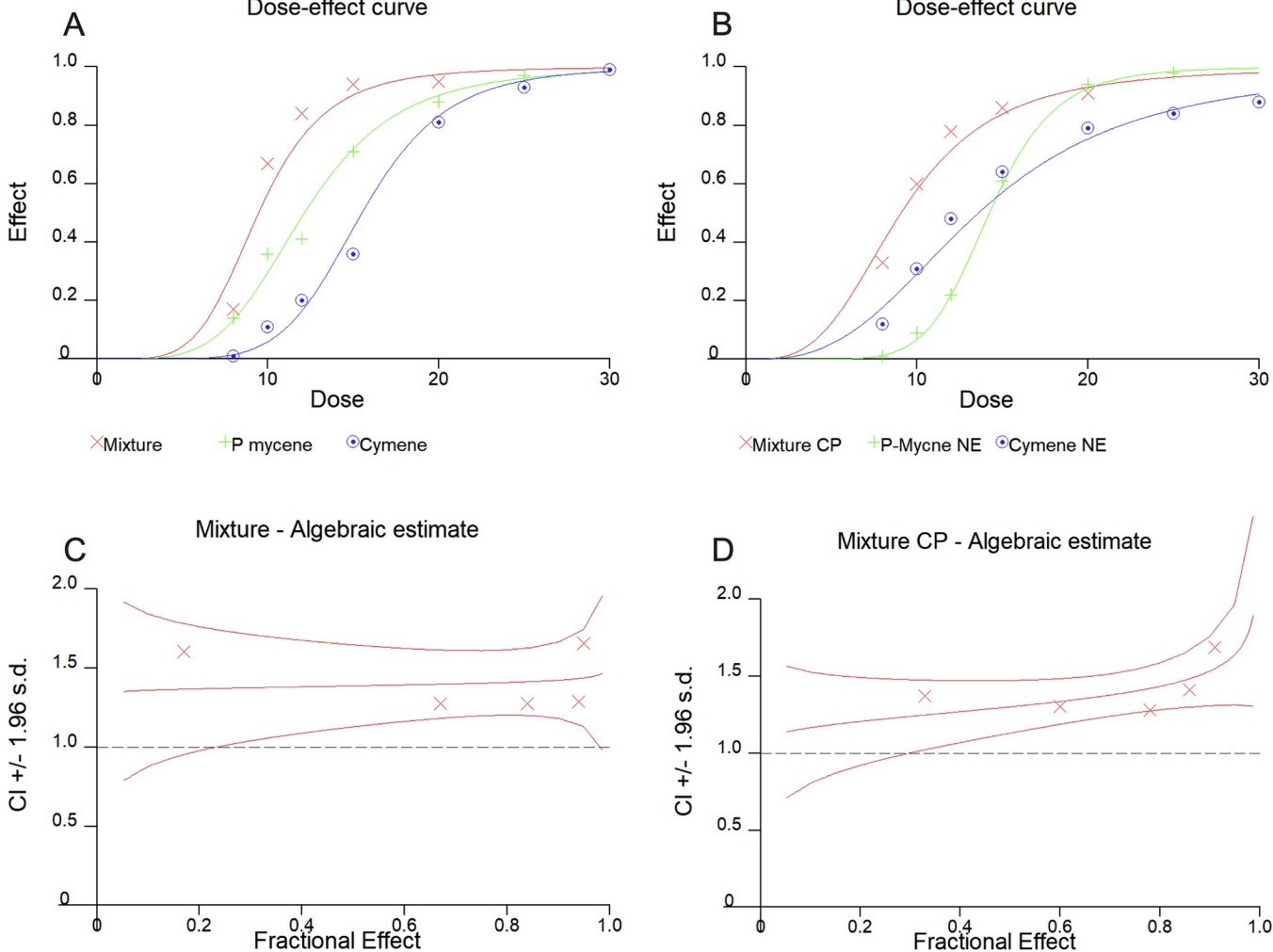

**Fig 2.** Concentrations effect curves of the combination of A—Myrcene:p-cymene (1:1) and B -Myr-NE:Cym-NE (1:1) and Interaction curves between C—Myrcene:p-cymene (1:1) and D -Myr-NE:Cym-NE (1:1).

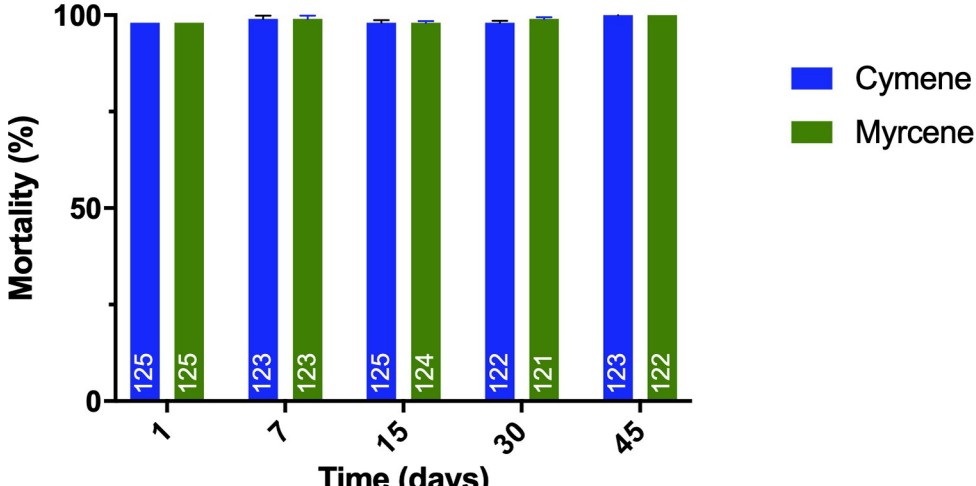

**Fig 3. Residual activity of Cym-NE and Myr-NE against susceptible larvae of *Ae. Aegypti*. numbers in the bars represent the sample size tested.**

## Residual activity

The results showed that the TNEs remain effective (providing 100% mortality) up to 45 days of storage (Fig 3), under laboratory-controlled conditions of temperature (27°C) and relative humidity (80%).

## Toxicity of terpenes against non-target organism

The treatment of zebrafish with Cym-NE (1750 and 2000 mg/kg) and Myr-NE (2000 mg/kg) triggered behavioral changes in zebrafish as shown in Table 4. However, mortality was observed in the group of animals treated at the dose of 2000 mg/kg of Ne-Cym. The signs of stress were an increase in swimming activity, tremors in the tail, loss of posture and motility, circular swimming and permanence at the bottom of the aquarium.

The main alterations observed in the tissues were: dilation of presents vessels in villi and hypertrophy of caliciform cells in the intestine, cytoplasmic vacuolization and nuclear atypia in the hepatocytes, prevalence of tubular alterations, such as mild tubular hyaline degeneration in the kidneys (Fig 4). The IHC calculated was less than 10, which classifies these organs as normal, because the alterations recorded were not able to alter the normal functioning of the organs.

**Table 4. Effect of oral treatment with Cym-NE and Myr-NE and control (water) over zebrafish behavior changes assessed in three stages.**

| Group | Doses | Stage I | Stage II | Stage III | Total | % |
|---|---|---|---|---|---|---|
| Control (water) | 2µl | 1/3 | 0/2 | 0/4 | 1/9 | 11.1 |
| Cym-NE | 1750 mg/kg | 2/3 | 2/2 | 2/4 | 6/9 | 66.6 |
| | 2000 mg/kg | 1/3 | 2/2 | 1/4 | 4/9 | 44.4 |
| Myr-NE | 2000 mg/kg | 2/3 | 2/2 | 2/4 | 6/9 | 66.6 |

Stage I: increase in swimming activity; Spasms; tremors in the tail; Stage II: circular swimming; loss of posture; Stage III: clonus; loss of motility; animal still at the bottom of the aquarium and death

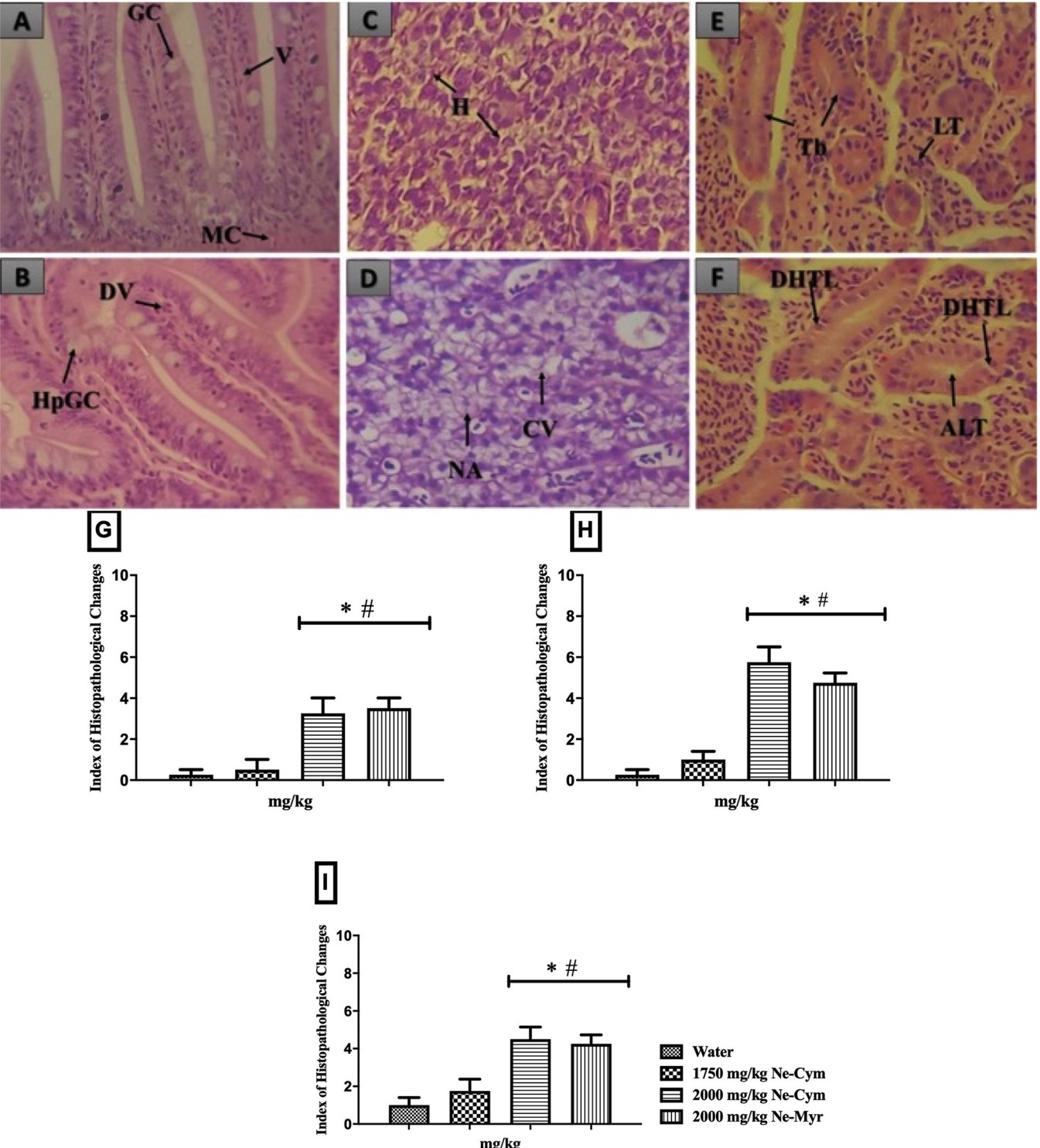

**Fig 4.** Histopathological changes observed in the intestine (A and B), liver (Cand D) and kidneys (E and F). In A and B intestinal tissue with caliciform cells (CC), villi (V), Muscle layer (ML), villi degeneration (VD) and caliciform cell hyperplasia (CCHP); In C and D normal hepatocytes (H), Cytoplasmatic vacuolization (CV), nuclear atrophy (NA); In E and F tubules (Tb), Lymphoid tissue (LT), Increase in tubular lumen (ITL), mild tubular hyaline degeneration (THD). H&E staining, IHC in the intestine (G), liver (H) and kidneys (I) of adult zebrafish in the acute oral toxicity test performed using Ne-Cym (1750 e 2000 mg/kg) e Ne-Myr (2000 mg/kg). Data show the mean ± SD (n = 5/group). Statistical analysis was performed through one-way ANOVA followed by the post hoc Tukey test.

## Discussion

In this study, we investigated the larvicidal activity of myrcene and *p*-cymene against the dengue vector *Ae. aegypti* and their potential toxicity against non-target organisms.

Our findings showed that myrcene and p-cymene exhibited moderate insecticidal activity against *Ae. aegypti* larvae when compared to the reference product Temephos. The lower efficacy of terpenes against mosquitoes has been reported previously [53, 54]. For example, Lee and Ahn (2013) reported that myrcene had a $LC_{50}$ of 36 mg/L, against third-instar larvae which is even higher than the $LC_{50}$ reported in the present study [53]. In contrast, Lucia et al. (2017) reported a lower $LC_{50}$ (12.49 mg/L) for p-cymene against third instar larvae [55]. These findings suggest that the potency of these compounds as mosquito control agents varies according to the studies and several factors such as the source and composition of the monoterpenes, experimental conditions, and mosquito species probably impact on the outcomes. Although TNEs show moderate larvicidal activity, our findings showed an absence of cross-resistance with commonly used pesticides and this offers interesting prospects for vector control considering that mosquito resistance is becoming an increasing threat to the prevention and control of mosquito borne diseases [8, 56]. Indeed, there's very limited number of insecticides having a different mode of action than the big "four" classes (i.e. pyrethroids, carbamates, organophosphate, organochlorines) and TNEs could then enrich the panel of vector control products used as part of integrated vector management.

Our study also compared the efficacy of free terpenes *versus* terpene-based nanoemulsions against *Ae. aegypti* mosquitoes. Our findings showed that nanoemulsions did not increase the efficacy of both myrcene and p-cymene against mosquito larvae. These results contrast with other findings showing an enhanced insecticidal activity of nanoemulsified terpenes or essential oils compared to non-encapsulated components [57, 58]. For example, the studies by Almadiy and Nenaah (2022) and Ferreira et al. (2020) used different essential oils and different particle sizes for their nanoemulsions. Alamdy used *Origanum vulgare* essential oil while Ferreira et al. (2020) used the Essential Oil of *Siparuna guianensis*. Additionally, the particle size of the nanoemulsions in Alamdy's study was 60 nm while Ferreira et al. reported a particle size of around 160 nm. Other strategies than nanoemulsions were used for the stabilization of essential oils, for example using yeast for the encapsulation of orange oil, and this strategy showed to be highly active ($LD_{50} < 50$ mg.$L^{-1}$) against *Ae. aegypti* [59, 60]. These differences in the chemical composition of the essential oils and the type of the stabilization method can affect the interaction of the terpenes or essential one's with mosquitoes and explain the different outcomes observed in our study compared to previous one's.

Despite that, our study showed a residual efficacy of the nanoemulsions for at least 45 days in a climatic chamber at constant temperature and humidity. This suggests that the use of nanoemulsions may prevent the volatilization of terpenoids and provide long-lasting activity as previously reported [61]. However, one bias of our study is that we couldn't compare the residual activity of the free terpenes *versus* encapsulated terpenes, due to a shortage of mosquitoes Moreover, it is important to highlight that laboratory conditions do not always reflect the field situation as external factors such as sunlight, rain, pollutants, etc can strongly reduce the residual efficacy of larvicides [62]. Further investigations are needed to assess the longevity of terpenes and TNEs in both laboratory and simulated field conditions.

Our research also investigated whether a combination of p-cymene and myrcene may be more effective at killing mosquitoes than each molecule used alone. The interest in using a combination of insecticides results in the fact that it may be possible to achieve better control of a mosquito population while reducing the doses of each compound, hence making this an environmentally friendly and cost-effective solution [45]. Our findings however showed that

the combination of terpenes did not kill a higher proportion of mosquitoes than the sum of the 2 terpenes taken separately, indicating additive or even slight antagonism at high concentrations. These findings contrast with other studies that have reported additive to synergistic interactions between various monoterpenes (e.g. terpinene, limonene, carvacrol, anethole, peroxide ascaridole, p-cymene and methyl isoeugenol) against mosquito larvae [63, 64]. It appears that the methodology used to assess the synergy and the compositions of terpenoid combinations differ between studies and results cannot be directly compared. In our study, the lack of synergistic interactions between p-cymene and myrcene might be explained by the fact that both terpenes act on the same target sites (i.e. mutually exclusive inhibitors) hence causing competitive inhibition for the same receptor, especially when concentrations increased (saturation of the system). More work is needed to better understand the interactions and mode of action of terpenoids in insects in order to select best active ingredients in combination.

Finally, we investigated the potential toxicity of the p-cymene and myrcene nanoemulsions against non-target organisms (fish) considering that these molecules may be potentially deployed in mosquito breeding habitats. Our results showed that TNEs lead to behavioral and histological changes in the Zebrafish at high concentrations. These alterations may lead to changes in body weight, food consumption, behavior, blood circulation, and reproductive functions [65]. Everds et al. (2013) state however that in toxicity trials, it is common to observe signs of stress in animals [66] and these histological alterations are considered normal, and not able to alter the normal functioning of the fish organs [67, 68]. Although it is unlikely that p-cymene and myrcene pose a significant hazard to non-target organisms, there is a need to assess their potential toxicity in simulated field conditions using the doses that may be applied in the field.

## Conclusions

Overall, our findings showed that TNEs have potential for vector control particularly as part of integrated vector management with the scope to preserve the susceptibility of mosquito populations to public health insecticides. Our findings suggest that the incorporation of monoterpenes into nanoemulsions is a promising approach to improve the solubility of these substances in aqueous media without organic solvents, which reinforces the vector control potential of these molecules.

## Acknowledgments

We thank all team from IRD-MIVEGEC for their support during the biological tests.

## Author Contributions

**Conceptualization:** Jonatas Lobato Duarte, Vincent Corbel.

**Formal analysis:** Jonatas Lobato Duarte, Stéphane Duchon, Leonardo Delello Di Filippo, Vincent Corbel.

**Funding acquisition:** Vincent Corbel.

**Investigation:** Jonatas Lobato Duarte, Stéphane Duchon.

**Resources:** Vincent Corbel.

**Supervision:** Marlus Chorilli, Vincent Corbel.

**Writing – original draft:** Jonatas Lobato Duarte, Leonardo Delello Di Filippo.

**Writing – review & editing:** Marlus Chorilli, Vincent Corbel.

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
