## [Decision Letter · Decision Letter 0]

26 Jun 2023

PONE-D-23-16634Larvicidal properties of terpenoid-based nanoemulsions against the Dengue vector Aedes aegypti L. and their potential toxicity against non-target organismPLOS ONE

Dear Dr. Corbel,

Thank you for submitting your manuscript to PLOS ONE. After careful consideration, we feel that it has merit but does not fully meet PLOS ONE’s publication criteria as it currently stands. Therefore, we invite you to submit a revised version of the manuscript that addresses the points raised during the review process.

We look forward to receiving your revised manuscript.

Kind regards,

Mozaniel Santana de Oliveira, Ph.D

Academic Editor

PLOS ONE

Journal Requirements:

"This project has received funding from Sao Paulo research foundation-FAPESP under the grants n°2021/11487-4 and 2019/25125-7. This project also receives a co-funding from the European Union HORIZON-MSCA-2021-SE-01 (INOVEC project), under the grant n°101086257. "

3. Please expand the acronym “FAPESP”  and "MSCA" (as indicated in your financial disclosure) so that it states the name of your funders in full.

Reviewers' comments:

Reviewer's Responses to Questions

**Comments to the Author**

1. Is the manuscript technically sound, and do the data support the conclusions?

Reviewer #1: Yes

Reviewer #2: Yes

2. Has the statistical analysis been performed appropriately and rigorously? 

Reviewer #1: Yes

Reviewer #2: No

3. Have the authors made all data underlying the findings in their manuscript fully available?

Reviewer #1: Yes

Reviewer #2: No

4. Is the manuscript presented in an intelligible fashion and written in standard English?

Reviewer #1: Yes

Reviewer #2: Yes

5. Review Comments to the Author

Reviewer #1: Good manuscript and well written. The concept is good as alternative method in insect control technology. The study has detailed view and experiments in terms of proving their concepts. Please refer to my comments for further improvement of this manuscripts.

Reviewer #2: The manuscript by Duarte et al. brings a straightforward presentation of the effects of two terpenoids and their nanoemulsions against Aedes aegypti larvae, and the test against a non-target organism. It is a well conducted and objective study. However, the presentation of data is very poor. I included some comments in the PDF file, but basically the main problems are:

1. Legends of Tables and figures are almost non-existant. There is no description of the meaning of variables, measurements, statistical tests and symbols, experimental designs. Please provide complete Legends.

2. Statistical analysis was performed without adequate report of results. Fos example, in some cases ANOVA was performed without reference to normality of data sets, which I think it is hard to demonstrate with N=5. Please provide proper statistical analysis.

6. PLOS authors have the option to publish the peer review history of their article (what does this mean?). If published, this will include your full peer review and any attached files.

Reviewer #1: **Yes: **Rajiv Ravi (PhD), Sumitomo Chemical Enviro-Agro Pacific Sdn.Bhd

Reviewer #2: No

---

## [Author Response · Author response to Decision Letter 0]

7 Jul 2023

We would like to express our sincere gratitude for the valuable feedback provided by the academic editor and reviewer(s) regarding our manuscript entitled " Larvicidal properties of terpenoid-based nanoemulsions against the Dengue vector Aedes aegypti L. and their potential toxicity against non-target organism." We appreciate their time and effort in thoroughly assessing our work. In response to their insightful comments, we have carefully considered each point raised and offer our rebuttal and revisions as follows:

REVIEWER #1

Something seems wrong here. Why the total is not 100%?

A: We corrected the information in the main text. 

What was the light exposure?

A: We used a light/dark cycle of 12 hours. We added this information in the methods section.

What is the meaning of AHI?

A: We corrected the translation mistake in the main text. The correct acronym is IHC. 

One way ANOVA with N=5 do not seem reasonable. how the authors tested the normality of distribution?

A: We used the Shapiro wilk test.

Decide between BORA or Bora and correct it along the manuscript.

A: Thank you for the suggestion. We standardized “Bora” throughout the text. 

Where is this data? refer it in the text please.

A: We included a reference to this data in the text. 

The legend of this Table is surprisingly poor. It´s not possible to know what p (italics and lowercase, please) means, or which statistical test was performed, or even the number of biological replicas (n is number of mosquitoes?). Please provide explanation for this data and a better statistical report (use addiditonal tables if necessary).

A: We reformulated the legend, adding the necessary information for data interpretation. 

Which test was used for the normality of distribution? ANOVA implies this condition.

A: We used the Shapiro wilk test to check the normality of the data.

Why GC?

A: We corrected the translation mistake in the main text.

Other strategies of stabilization of essential oils by encapsulation were alerady proposed, for example yeast encapsulation, as demonstrated in: 

1. Parasit Vectors. 2021 May 22;14(1):272. doi: 10.1186/s13071-021-04733-2.

2. Parasit Vectors. 2020 Jan 13;13(1):19. doi: 10.1186/s13071-019-3870-4.

A: We appreciate the suggestion. We would like to reiterate that the objective of this article is to discuss approaches related to nanoemulsions and larvicidal activity. At this time, we have chosen not to include other approaches in the manuscript.

REVIEWER #2

Introduction

Good explanations on the introduction about the mechanisms of resistant and objective of this study. Very clear and on the points for a research statements. The experiment design is much targeted and design well. Appropriate in this context. 

Line 56-65 can be mentioned more comprehensive, especially the alternative method. You are not using those method and not relevant in your study scope

A: Thank you for the suggestion. We have chosen to remove that section from the manuscript.

Explain more on the Nano emulsions technology for slow release. How the mechanisms work for slow release for a highly volatile component. Line 92-96

A: We appreciate the reviewer's comment. We have included a detailed explanation on Nano emulsion technology for slow release in the manuscript. 

Materials and Method

If you have pictures of behavioral analysis and mortality of fish, please include. This will be a good reference for repeating this kind of experiment.

A: Thank you for your suggestion. While we understand the value of including pictures of behavioral analysis and fish mortality in the manuscript, we would like to clarify that these specific visuals are not available in our study. However, we have provided detailed descriptions of the behavioral analysis and fish mortality results in the text. Additionally, we have referenced relevant literature that provides visual documentation of similar experiments, which can serve as a valuable reference for researchers interested in replicating these experiments.

Explain the line 186-193 with the picture for each Stage I-III

A:Thank you for your suggestion. While we understand the value of including pictures of behavioral analysis and fish mortality in the manuscript, we would like to clarify that these specific visuals are not available in our study.

Results

Table 2 and Table 3, please include in title the RR ratio index range based on WHO reference. Example; According to WHO RR < 5 indicates a susceptible field population; RR from 5-10 indicates moderate resistance and RR > 10 indicates high resistance, (Monitoring and managing insecticide resistance in Aedes mosquito populations. Interim guidance for entomologists. WHO/ZIKV/VC/16.1. Geneva: World Health Organization; 2016)

A: We included this information in the method section as this is not a result (P148-150).

We hope that these revisions adequately address the concerns raised and align the manuscript more closely with the objectives and scope of PLOS ONE. We are confident that the revised version now meets the high standards of academic excellence set by your esteemed journal.

Once again, we would like to express our sincere appreciation for the thorough review and constructive feedback provided by the academic editor and reviewer(s). Their expertise and guidance have undoubtedly improved the quality of our work.

Thank you for considering our revised manuscript. We look forward to receiving your final decision regarding its publication.

Yours sincerely,

Vincent Corbel

---

## [Decision Letter · Decision Letter 1]

31 Jul 2023

PONE-D-23-16634R1

Larvicidal properties of terpenoid-based nanoemulsions against the Dengue vector Aedes aegypti L. and their potential toxicity against non-target organism

PLOS ONE

Dear Dr. Corbel,

Thank you for submitting your manuscript to PLOS ONE. After careful consideration, we have decided that your manuscript does not meet our criteria for publication and must therefore be rejected.

I am sorry that we cannot be more positive on this occasion, but hope that you appreciate the reasons for this decision.

Kind regards,

Mozaniel Santana de Oliveira, Ph.D

Academic Editor

PLOS ONE

Reviewers' comments:

Reviewer's Responses to Questions

**Comments to the Author**

1. If the authors have adequately addressed your comments raised in a previous round of review and you feel that this manuscript is now acceptable for publication, you may indicate that here to bypass the “Comments to the Author” section, enter your conflict of interest statement in the “Confidential to Editor” section, and submit your "Accept" recommendation.

Reviewer #1: All comments have been addressed

Reviewer #3: (No Response)

2. Is the manuscript technically sound, and do the data support the conclusions?

Reviewer #1: Yes

Reviewer #3: (No Response)

3. Has the statistical analysis been performed appropriately and rigorously? 

Reviewer #1: Yes

Reviewer #3: (No Response)

4. Have the authors made all data underlying the findings in their manuscript fully available?

Reviewer #1: Yes

Reviewer #3: (No Response)

5. Is the manuscript presented in an intelligible fashion and written in standard English?

Reviewer #1: Yes

Reviewer #3: (No Response)

6. Review Comments to the Author

Reviewer #1: 1. All the questions has been answered and acceptable for publications

2. Format is acceptable

3. Manuscript is acceptable

Reviewer #3: The authors did not make the necessary changes for the manuscript to be published in this journal.

7. PLOS authors have the option to publish the peer review history of their article (what does this mean?). If published, this will include your full peer review and any attached files.

Reviewer #1: **Yes: **Rajiv Ravi

Reviewer #3: No

- - - - -

---

## [Author Response · Author response to Decision Letter 1]

17 Aug 2023

We would like to express our sincere gratitude for the valuable feedback provided by the academic editor and reviewer(s) regarding our manuscript entitled " Larvicidal properties of terpenoid-based nanoemulsions against the Dengue vector Aedes aegypti L. and their potential toxicity against non-target organism." We appreciate their time and effort in thoroughly assessing our work. In response to their insightful comments, we have carefully considered each point raised and offer our rebuttal and revisions as follows:

REVIEWER #1

Something seems wrong here. Why the total is not 100%?

A: We corrected the information in the main text. 

What was the light exposure?

A: We used a light/dark cycle of 12 hours. We added this information in the methods section.

What is the meaning of AHI?

A: The correct acronym is IHC (Index of Histopathological Changes). We corrected the mistake in the text. 

One way ANOVA with N=5 do not seem reasonable. how the authors tested the normality of distribution?

A: We used the Shapiro wilk test to check the distribution. The data showed normal distribution (Statistic 0.9598, P value: 0.6577).

Decide between BORA or Bora and correct it along the manuscript.

A: Thank you for the suggestion. We standardized “Bora” throughout the text. 

Where is this data? refer it in the text please.

A: We included a reference to this data in the text. 

The legend of this Table is surprisingly poor. It´s not possible to know what p (italics and lowercase, please) means, or which statistical test was performed, or even the number of biological replicas (n is number of mosquitoes?). Please provide explanation for this data and a better statistical report (use addiditonal tables if necessary).

A: We reformulated the legend, adding the necessary information for data interpretation. 

Which test was used for the normality of distribution? ANOVA implies this condition.

A: We used the Shapiro wilk test to check the normality of the data. The data showed normal distribution (Statistic 0.9598, P value: 0.6577).

Why GC?

A: We corrected the translation mistake in the main text.

Other strategies of stabilization of essential oils by encapsulation were already proposed, for example yeast encapsulation, as demonstrated in: 

1. Parasit Vectors. 2021 May 22;14(1):272. doi: 10.1186/s13071-021-04733-2.

2. Parasit Vectors. 2020 Jan 13;13(1):19. doi: 10.1186/s13071-019-3870-4.

A: We appreciate the suggestion. We inserted the suggested sentence and references in the manuscript.

REVIEWER #2

Introduction

Good explanations on the introduction about the mechanisms of resistant and objective of this study. Very clear and on the points for research statements. The experiment design is much targeted and design well. Appropriate in this context. 

Line 56-65 can be mentioned more comprehensive, especially the alternative method. You are not using those method and not relevant in your study scope

A: Thank you for the suggestion. We have chosen to remove that section from the manuscript.

Explain more on the Nano emulsions technology for slow release. How the mechanisms work for slow release for a highly volatile component. Line 92-96

A: We appreciate the reviewer's comment. We have included a detailed explanation on Nano emulsion technology for slow release in the manuscript (see introduction). 

Materials and Method

If you have pictures of behavioral analysis and mortality of fish, please include. This will be a good reference for repeating this kind of experiment.

A: Thank you for your suggestion. While we understand the value of including pictures of behavioral analysis and fish mortality in the manuscript, we would like to clarify that these specific visuals are not available in our study. However, we have provided detailed descriptions of the behavioral analysis and fish mortality results in the text. Additionally, we have referenced relevant literature that provides visual documentation of similar experiments, which can serve as a valuable reference for researchers interested in replicating these experiments.

Explain the line 186-193 with the picture for each Stage I-III

A:Thank you for your suggestion. While we understand the value of including pictures of behavioral analysis and fish mortality in the manuscript, we would like to clarify that these specific visuals are not available in our study.

Results

Table 2 and Table 3, please include in title the RR ratio index range based on WHO reference. Example; According to WHO RR < 5 indicates a susceptible field population; RR from 5-10 indicates moderate resistance and RR > 10 indicates high resistance, (Monitoring and managing insecticide resistance in Aedes mosquito populations. Interim guidance for entomologists. WHO/ZIKV/VC/16.1. Geneva: World Health Organization; 2016)

A: We included this information in the method section as this is not a result (P148-150).

We hope that these revisions adequately address the concerns raised and align the manuscript more closely with the objectives and scope of PLOS ONE. We are confident that the revised version now meets the high standards of academic excellence set by your esteemed journal.

Once again, we would like to express our sincere appreciation for the thorough review and constructive feedback provided by the academic editor and reviewer(s). Their expertise and guidance have undoubtedly improved the quality of our work.

Thank you for considering our revised manuscript. We look forward to receiving your final decision regarding its publication.

Yours sincerely,

Vincent Corbel

---

## [Editor Report · Decision Letter 2]

6 Oct 2023

Larvicidal properties of terpenoid-based nanoemulsions against the Dengue vector Aedes aegypti L. and their potential toxicity against non-target organism

PONE-D-23-16634R2

Dear Dr. Corbel,

We’re pleased to inform you that your manuscript has been judged scientifically suitable for publication and will be formally accepted for publication once it meets all outstanding technical requirements.

Kind regards,

Pedro L. Oliveira

Academic Editor

PLOS ONE
---

## [Editor Report · Acceptance letter]

11 Oct 2023

PONE-D-23-16634R2 

Larvicidal properties of terpenoid-based nanoemulsions against the Dengue vector *Aedes aegypti* L. and their potential toxicity against non-target organism 

Dear Dr. Corbel:

I'm pleased to inform you that your manuscript has been deemed suitable for publication in PLOS ONE. Congratulations! Your manuscript is now with our production department. 

Kind regards, 

on behalf of

Dr. Pedro L. Oliveira 

Academic Editor

PLOS ONE